# Community-Acquired *Clostridioides difficile* Infection: The Fox Among the Chickens

**DOI:** 10.3390/ijms26104716

**Published:** 2025-05-14

**Authors:** Panagiota Xaplanteri, Chrysanthi Oikonomopoulou, Chrysanthi Xini, Charalampos Potsios

**Affiliations:** 1Department of Microbiology, General Hospital of Eastern Achaia, 25100 Aigio, Greece; 2Department of Microbiology, General Hospital of Eastern Achaia, 25001 Kalavrita, Greece; xrysaoiko@gmail.com; 3Department of Microbiology, Attikon University General Hospital, 12462 Athens, Greece; chrysanthixini1984@gmail.com; 4Department of Internal Medicine, University General Hospital of Patras, 26504 Patras, Greece; charpotsios@gmail.com

**Keywords:** community-acquired *Clostridioides difficile* infection, food-borne infection, inflammatory bowel disease, immunosuppression, asymptomatic carriage

## Abstract

*Clostridioides difficile* infection (CDI) appears mainly as nosocomial antibiotic-associated diarrhea, and community-acquired infection is increasingly being recognized. The threshold of asymptomatic colonization and the clinical manifestation of CDI need further elucidation. Community-acquired CDI (CA-CDI) should be considered when the disease commences within 48 h of admission to hospital or more than 12 weeks after discharge. Although CDI is not established as a food-borne or zoonotic disease, some data support that direction. The spores’ ability to survive standard cooking procedures and on abiotic surfaces, the formation of biofilms, and their survival within biofilms of other bacteria render even a low number of spores capable of food contamination and spread. Adequate enumeration methods for detecting a low number of spores in food have not been developed. Primary care physicians should take CA-CDI into consideration in the differential diagnosis of diarrhea, as there is a thin line between colonization and infection. In patients diagnosed with inflammatory bowel disease and other comorbidities, *C. difficile* can be the cause of recurrent disease and should be included in the estimation of diarrhea and worsening colitis symptoms. In the community setting, it is difficult to distinguish asymptomatic carriage from true infection. For asymptomatic carriage, antibiotic therapy is not suggested but contact isolation and hand-washing practices are required. Primary healthcare providers should be vigilant and implement infection control policies for the prevention of *C. difficile* spread.

## 1. Introduction

*Clostridioides difficile* infection (CDI) appears mainly as nosocomial antibiotic-associated diarrhea, and community-acquired infection is increasingly being recognized [1]. The infection risk in patients colonized with the bacterium differs in the literature and depends on the healthcare setting [1]. The threshold of asymptomatic colonization and the clinical manifestation of CDI need further elucidation [2,3]. The onset of symptoms defines community-acquired CDI (CA-CDI). If the disease commences within 48 h of hospital admission or more than 12 weeks after discharge, it can be considered a non-nosocomial infection [4]. Recurrent CDI is described in 35% of the patients who suffer from the disease, rendering CDI a considerable threat [5]. *C. difficile* colonization and CA-CDI are of concern in patients diagnosed with inflammatory bowel disease (IBD) and other comorbidities [6,7,8]. This study aimed to delineate existing data regarding the spread of *C. difficile* in the community, both in healthy individuals with no predisposing factors and in patients suffering from immunosuppression and other underlying diseases. Distinction between colonization and CDI in the community setting is difficult. CDI and asymptomatic carriage require contact isolation and hand-washing practices, and primary healthcare providers should be vigilant to reduce spread [9]. The spores of the bacterium are highly infectious and survive alcohol-based disinfectants and on abiotic surfaces in both hospital and domestic environments. Patients suffering from CDI release up to 1 × 10^7^ spores per gram of feces [10].

## 2. Pathogenesis

*C. difficile* is a Gram-positive toxin-producing bacillus. It develops strictly in anaerobic conditions, forming aerotolerant, metabolically dormant endospores [1,11].

Its name derives from the difficulty of isolating the microorganism in culture [12].

It was first described as a fecal commensal of healthy newborns [13]. The use of broad-spectrum antibiotics from the 1960s onward has led to disease manifestations that vary from mild, self-limiting disease to life-threatening sepsis and death [1].

Since 2000, with the introduction of comparative genomics, cases of community-associated CDI in patients with no former risk factors associated with nosocomial colonization have been increasing, giving a different perspective on the disease’s epidemiology [11].

### 2.1. Virulence Factors of C. difficile

*Clostridioides difficile* releases various toxins that are considered major virulence factors of the bacterium [14]. Glucosyltransferases, large glycosylating exotoxin A, an enterotoxin (TcdA), and large glycosylating cytotoxin B (TcdB) are major virulent factors. TcdA in animal models causes diarrhea and disrupts the colonic mucosa [15]. The production of these toxins is controlled by genes present only in toxigenic strains. These genes are encoded in the 19 kb pathogenicity locus (PaLoc), which is rarely mobilized via homologous recombination. Acquisition of the PaLoc via horizontal gene transfer renders non-toxigenic strains toxigenic [1,12,16]. TcdB is related to severe localized colonic injury and systemic toxemia, associated with extraintestinal organ damage that leads to multiple organ dysfunction syndrome (MODS), a fatal complication of CDI [17]. In animal infection models, TcdB-producing strains are more virulent than the respective deficient isogenic mutants [17]. Aligned to these findings are clinical observations that TcdA-TcdB+ strains cause more severe disease [18].

The functional structure of both TcdA and TcdB comprises four structure domains with 48% homology: the N-terminal glucosyltransferase domain (GTD), the autoprocessing domain (APD), the translocation/pore-forming domain, and the C-terminal combined repetitive oligopeptide repeat (CROP)3 domain [19].

The C-terminal domain of toxins A and B acts as receptor recognition and a ligand on host cells [20]. TcdA binds to the receptor disaccharide Galβ1-4GlcNac to enter the host cell, leading to endocytosis [15]. The luminal aspect of the colonic epithelium has specific brush border binding receptors for the virulence factors of *C. difficile* toxins A and B.

Regarding TcdB, to our knowledge, the cell-surface proteins chondroitin sulfate proteoglycan 4 (CSPG4), poliovirus receptor like 3 (PVRL3, or NECTIN3), and members of the Frizzled protein family (FZD1, FZD2, and FZD7) have been identified as receptors of the toxin, but their role is still to be elucidated [19]. The ligand of CSPG4 is the combined repetitive oligopeptide (CROP) domain of TcdB, and CSPG4 is the only CROP-specific receptor for TcdB identified so far [19]. The first three short oligopeptide repeats from the CROPs are essential for CSPG4 binding and full cellular toxicity induced by the toxin [19].

NECTIN3 in the presence of high concentrations of TcdB is responsible for colonic cell death [19].

The toxin enters the intestinal epithelial cells via those receptors and adds a glucose moiety to a specific threonine on Rho proteins, leading to their inactivation. As a result, the intestinal cell loses its cytoskeletal architecture, actin filaments depolymerize, and cell death occurs. The toxin receptor density significantly affects the outcome and determines the disease severity [12]. TcdB causes also colonic cell necrosis via a non-glucosylation mechanism [21,22].

An additional way of entry into the cytosol is pore formation on host cells. This activity is encoded by a hydrophobic translocation domain on both toxins A and B [20]. Autocatalytic cleavage leads to the release of the active N-terminal glucosyltransferase domain into the host cell cytosol [20]. The N-terminal domain is highly conserved and responsible for substrate specificity [15]. Upon entry into the host cells, the N-terminal glucosyltransferase domains of both toxins inactivate the Ras superfamily of small GTPases, provoking irreversible alterations to vital cell-signaling pathways [15].

The toxins also act directly on the intestinal lamina propria, provoking ulcerations and triggering the formation of an inflammatory pseudomembrane [23]. These toxins lead to the characteristic clinical symptoms of CDI [12]. Therefore, the presence of toxins A or B in stools and the consequential cytotoxicity confirm the diagnosis of CDI. Tissue culture cytotoxicity assay is considered the gold standard for diagnosis but is expensive and time-consuming. Establishment of laboratory diagnosis with more rapid tests varies. There are antigen tests that detect the antigen glutamate dehydrogenase (GDH) of *C. difficile*. This test may identify the presence of the bacterium in stool, but further testing is required to prove toxigenicity and CDI [24]. Toxin antigen testing in stool is quick and gives results rapidly but has limitations. Toxins A and B are undetectable two hours after specimen collection, and improper specimen handling may lead to false negative results [24].

Tissue culture cytotoxicity assay has high specificity and sensitivity for toxin B, but it is time- and money-consuming [24]. Enzyme-linked immunosorbent assays (ELISAs) for *C. difficile* toxins are used in daily laboratory practice [12]. The test provides results within the same day and has a specificity of 99% but it lacks sensitivity in comparison to tissue culture cytotoxicity and PCR or toxigenic culture [24]. Stool cultures for *C. difficile* isolation alone do not discern toxigenic from non-toxigenic strains. Moreover, stool cultures need specific anaerobic conditions involving up to 96 h of cultivation, and isolation of the microorganism requires further testing for toxin production [24]. Molecular PCR assays have high specificity and sensitivity for detection of toxigenic strains, but results should be interpreted with caution in patients without symptoms [24].

In addition, endoscopy that reveals rectal pseudomembranes can be used in cases where diagnosis is in doubt. As the risk of perforation is evident, endoscopy should be performed with the introduction of minimum amounts of air and limited to the rectum or sigmoid colon in patients with antibiotic-induced diarrhea and the presence of pseudomembranes to establish a diagnosis [12].

Both TcdA and TcdB inactivate Rho, Rac, and Cdc42 within target cells, provoking actin cytoskeleton damage, the subsequent apoptosis of colonocytes, and the disruption of the tight junctions of epithelial barriers [15]. The altered integrity of tight junctions provokes increased intestinal permeability and diarrhea and triggers neutrophil accumulation. Neutrophils directly interact with TcdA, leading to the inflammatory cascade of pseudomembranous colitis. TcdA also recruits neutrophils, maintaining the inflammation [15,20]. Both toxins mediate profound proinflammatory responses via the activation of the inflammasome and TLR4 and TLR5 receptors. The cascade of intracellular signaling pathways that follow leads to the production of the interleukins IL-12, IL-18, IL-1b, interferon-g, and tumor necrosis factor-a (TNF-a) [20].

TcdB induces colonic cell death via upregulating Duox2 pathways which induce reactive superoxide species similarly to NADPH oxidase [22]. ROS are well described in the literature as causative pathogenetic agents of small intestine ischemia and ulcerative colitis [25,26,27].

A third toxin, the binary ADP-ribosylating toxin, or *C. difficile* transferase (CDT), has been reported in some strains related to more severe CDI disease [1,2]. These strains also include the hypervirulent epidemic strain NAP1/027. These hypervirulent strains are named type 027 via PCR ribotyping, group BI via restriction endonuclease analysis, and type NAP1 (027/BI/NAP1) via pulsed-field gel electrophoresis. The same strains also possess an 18 bp deletion and a stop codon in the *tcdC* gene that downregulates the production of toxins A and B [1,2]. Binary ADP-ribosylating toxin has two distinct components: CDTa and CDTb. CDTb binds lipolysis-stimulated lipoprotein receptors on the intestinal cells and promotes ADP ribosylation. CDTa acts on host cell actin filaments. Both components lead to the disruption of the host cell cytoskeleton and the creation of membrane protrusions that facilitate bacterial adherence [2]. Besides PCR ribotype 027, strains reported to cause severe disease include ribotypes 0126, 018, 056, 078, and 244 [14]. CDT has a similar action to *C. perfringens* iota toxin (i-toxin). CDTb is a ligand to a receptor found in cell surface enteric cells called lipolysis-stimulated lipoprotein receptor (LSR). The CDTb-LSR complex facilitates the binding and entering of CDTa into the cytosol [20]. CDTa ribosylates actin, which leads to the disruption of the cytoskeleton structure and cell death (apoptosis) [20]. Hybrid toxins have also been identified. Toxin TcdB-1470 of *C. difficile* strain 1470 has structural similarities to TcdB but can modify Ras, Rac, Rap, and Ral [15]. The *C. difficile* genome comprises up to 11% mobile genetic elements. These genetic elements provide antibiotic resistance and the modulation and transfer of toxin genes. Acquisition of these genes can render non-toxigenic strains toxin producers [16,28]. Transposable element Tn6218 is non-PaLoc-dependent and is related to significant antibiotic resistance. The *ermB* and *cfr* genes are related to clindamycin resistance [14]. Regarding biofilm formation, *C. difficile* can form biofilms that allow the microorganism to survive oxygen stress and disinfectants on abiotic surfaces in the nosocomial setting or domestic and food industry environments. *C. difficile* spores can also survive within biofilms formed from other bacteria [28,29,30,31,32,33]. *C. difficile* spores are the form of survival in an unfriendly milieu and the infectious particles of the bacterium. They are difficult to destroy with the usual methods used in healthcare facilities. Relapses of CDI can be due to persistence of the spores of the bacterium in the gut via biofilm formation [34]. Biofilm shields the vegetative cells of the bacterium against antibiotics and is a reservoir of spores [35]. Although there are many studies in the literature, biofilm formation by *C. difficile* needs further elucidation [36]. The bacterial genome within the biofilm is expressed 20% differently in comparison to planktonic cells [35]. Genes related to *C. difficile* biofilm formation are the *dnaK gene*, *lexA*, *spo0A*, quorum sensing regulator *luxS*, and germination receptor *sleC* [36]. Excess succinate is another strong inducer of *C. difficile* biofilm formation. In a healthy gut environment, succinate production is regulated by gut microflora. Dysbiosis disrupts this balance. Succinate is involved in host bacterial clearance via induction of high levels of inflammatory cytokine IL-1β by inhibiting the negative regulator of hypoxia-induced factor 1α (HIF-1α) [34]. Extracellular succinate induces the formation of thicker and more complex architecture biofilms through mechanisms that involve major metabolic shifts and cell-wall composition changes [34]. *C. difficile* type IV pili (T4P) are important in early biofilm formation [37]. Spo0A, a master regulator for *C. difficile* sporulation, is involved in early biofilm formation [29,30,32,38,39]. In mice, the *C. difficile* Spo0A protein is the protagonist in disease persistence and recurrence [39]. The presence of sub-lethal concentrations of the bile salt deoxycholate (DOC) triggers *C. difficile* biofilm formation. Sub-lethal concentrations of DOC are present after antibiotic therapy for CDI in the context of gut microbiota balance restoration, favoring recurrent infection [40]. *C. difficile* cells in biofilms show increased production of fatty acids related to the *lcpB* gene [36]. DNA derived from cell lysis, extracellular DNA (eDNA), is a major component of *C. difficile* biofilm formation [36]. In vitro, in the early stage of biofilm formation, some planktonic cells undergo autolysis to release eDNA [35]. This process is mediated by cell-wall proteins like Cwp19, which hydrolyses peptidoglycan and toxin–antitoxin expression systems [35]. Another mechanism described in the literature that favors eDNA release is phage-mediated cell lysis [41]. A well-described trigger for *C. difficile* biofilm formation is exposure to vancomycin and metronidazole and hydrophobicity due to the glycan of the bacterium flagella [29,42]. Flagella of the bacterium are required in vitro in the late stages of biofilm formation [31]. Within *C. difficile* biofilms, synthesis of diguanylate cyclase is upregulated. Diguanylate cyclase triggers the production of cyclic di-30,50-guanylate (cyclic-di-GMP), a second messenger molecule which is involved in the posttranscriptional regulation of biofilm formation [35,43]. Cyclic-di-GMP is the cornerstone of transition to a biofilm state. It is also involved in suppression of flagellar motility and induction of type IV pili production [44]. Cyclic-di-GMP also regulates cell-wall binding protein (Cwp11), cell surface protein (Cwp10), and calcium-binding adhesion protein [43,44]. Cwp11 is secreted during biofilm formation and cwp10 is increasingly expressed in 630Δerm biofilms in comparison to planktonic cells [43]. Interspecies LuxSCD and intraspecies accessory gene regulator (Agr) are considered the major quorum sensing systems in *C. difficile* biofilms [36]. The LuxS QS system uses the signaling molecule autoinducer-2 (AI-2) to generate an intracellular signaling cascade that leads to gene regulation [43]. *C. difficile* LuxS/AI-2 is a key component in the formation of single- and multi-species communities within which the bacterium survives. In addition, LuxS induces *C. difficile* prophages, favoring eDNA release and reinforcement of the biofilm scaffold [41]. Increased expression of the *agrD1 gene* in *C. difficile* biofilms has been reported in vitro [35]. In experimental models, *C. difficile* cells survive within colonic microbiota biofilms, which act as a protective niche against vancomycin and fecal microbiota transplant therapies [45,46].

### 2.2. Means and Sources of Colonization

The ingested bacterium first confronts the natural barriers of the innate immune system. *C. difficile*’s first encounter is gastric acidity, which can minimize the number of viable bacteria and inactivates toxins A and B [12]. Factors that influence gastric pH, such as long-term intake of proton-pump inhibitors, may facilitate the bacterium’s survival. In the literature, a gastric pH above 5.0 leads to direct intestinal colonization. Proton-pump inhibitors disrupt the balance of *Firmicutes* in favor of *Bacteroidetes*, which acts in favor of CDI [14]. Colonization of the bacterium presupposes an imbalance of the normal colonic commensals and patients’ exposure to an environment abundant with the microorganism or its spores. The ideal environment described so far is the nosocomial setting [47]. The reservoirs of the infection within hospital settings are the asymptomatic carriers. The bacterium’s spores are antibiotic-resistant and survive on all surfaces and the personnel’s hands, facilitating environmental contamination and patient-to-patient spread [12].

*C. difficile* can inhabit all the natural milieu, including fresh- and seawater and soil. Recent data in the literature show that food products could be a vehicle for the ingestion of *C. difficile* spores and subsequent infection in the community [48,49,50,51].

The bacterium’s spores can contaminate food products in nature and domestic environments. *C. difficile*’s capacity to provoke food-borne disease still needs further investigation [14]. The spores of *C. difficile* in nature can be found in soil, fresh- and seawater, wastewater from treatment plants discharged into potable water sources, and seafood. Composted manure as soil fertilizer can contain *C. difficile* spores that are transferred to fresh vegetables and fruits [52,53,54].

Meat can bear *C. difficile* spores acquired in the slaughterhouse environment and transmitted to humans via the food industry and food chain [55]. In the literature, *C. difficile* strains reported from patients with community-associated CDI, such as PCR ribotypes 017, 027, and 078, have been isolated from food products. In Europe, the extremely virulent strain ribotype 078 has been isolated from pork, beef, and mussels [14]. Food vacuum packaging, especially in ready-to-eat products, where all the oxygen is removed, facilitates the bacterium’s survival [56]. In addition, the bacterium’s spores are resistant to usual cooking temperatures. The endospores’ capacity to survive in extreme temperatures and alcohol renders colonization via ingestion a possibility even in populations with no risk factors related to antibiotic use or hospitalization. Therefore, ready-to-eat or undercooked food may contain spores of toxigenic strains and could be the contamination route [56]. The bacterium’s spores have been identified on kitchen food preparation surfaces and inside refrigerators in the domestic environment [57]. The survival of the bacterium’s spores from many disinfectants favors the contamination of kitchen surfaces and refrigerators [58].

Soil fertilizers derived from composted horse and pig manures can contain the microbe [59]. The spores of *C. difficile* can be transferred to the food chain via this route by consuming raw or undercooked vegetables and fruits [52]. The feces of carriers or infected livestock with the bacterium can transmit the spores to the meat [55]. Strains isolated mainly from cattle, such as ST11, are responsible for more severe disease [60]. In China, the same strain has been isolated from pigs [61]. Viscera of the animal carriers at slaughterhouses, especially the gut content that contains the spores, favor the contamination of meat [55]. The bacterium’s spores have been found in beef, pork, and poultry in retail markets [57,62,63,64,65,66,67,68].

Ready-to-eat meals and products with the oxygen removal technique during packaging benefit the bacterium’s survival [56]. Additionally, *C. difficile* can survive in the presence of food preservatives, such as nitrite, nitrate, and sodium metabisulfite, as applied in the food industry. The maximum concentration permitted by law for food preservatives in ready-to-eat products, such as sodium nitrite (E250), sodium nitrate (E251) and sodium metabisulfite (E223), is inadequate to prevent germination of spores [69,70].

Regarding cooking and standard food processing, the bacterium’s spores are resistant to freezing for up to 12 weeks in ground beef. A cooking temperature of 71 °C does not destroy the spores. These characteristics favor CDI even when low levels of spores are present in food [71,72].

No gold standard or ISO procedures have been developed for detecting *C. difficile* in raw materials and food products in the food industry. The spores’ ability to survive standard cooking procedures and on abiotic surfaces, the formation of biofilms, and their survival within biofilms of other bacteria render even a low number of spores capable of food contamination and spread. There is no unique method for detecting and enumerating *C. difficile* in food with accuracy [69]. Another disadvantage in enumeration and detection methodologies applied so far is that a food product can be tested falsely negative, as *C. difficile* distribution in food is not homogeneous [57]. Therefore, the most universally accepted method of detection is isolation of the microorganism in animal stool samples and toxin production investigation [69].

Ingestion of the vegetative form or preformed toxins in healthy individuals is a less likely scenario since they cannot survive the acidic stomach pH. Meanwhile, susceptible patients could be at risk for CDI [73]. The data in the literature regarding the use of proton-pump inhibitors (PPIs), the intake of which augments the gastric pH, are conflicting and depend on the underlying illness [73].

## 3. Community Spread in Healthy Individuals and Immunosuppressed Patients—The Distinct Role of Gut Microbiota

Individuals exposed to healthcare facilities can become asymptomatic carriers [74]. Hospitalized patients who become carriers of the bacterium are seven to one compared with those who develop the disease. These patients can spread the bacterium in the community [74]. Carers of infants are also at risk, as it is well-known that infants can be asymptomatic carriers of *C. difficile* [75].

Case reports and population-based studies of community-acquired infection have been described in the literature. In these studies, patients with no underlying health issues or no former antibiotic administration 12 weeks before disease development suffered from CDI [74,76,77,78,79]. Although patients with CA-CDI may be younger with less comorbidities, older age and antibiotic-induced dysbiosis remain significant factors for acquisition of true infection. This is one of the reasons why it is important to differentiate colonization from true infection, especially in the community setting, as it is likely that a significant portion of these “patients” are actually not infected.

The community spread of nosocomial strains is evident since strains with higher toxigenicity and drug resistance have been isolated [80]. In other studies, there was no dominant strain in the community-acquired disease [81].

A healthy intestinal environment in combination with gut peristalsis prohibits the colonization of *C. difficile*. Other strictly anaerobic bacteria of normal flora such as *Bacteroides* species act competitively and prevent colonization in humans above two years old [12,82]. A balanced gut microbiota is the key component to avoid the germination of the endospores in individuals who have ingested the bacterium’s spores. The microflora is responsible for the metabolization of primary bile acids that favor the germination of the spores to secondary bile acids that have the opposite effect. Thus, an imbalance of the gut microbiota facilitates the germination and growth of the bacterium and the subsequent toxin production of the toxigenic strains [11].

Healthy gut microbiota inhibits spore germination in the lower ileum, where the secondary bile acids naturally prevail. The disruption of the equilibrium that favors the primary bile acids in the area favors germination and toxin production [11].

The germination process of ingested spores takes place in the small intestine. The toxin production of vegetative forms commences in the anaerobic milieu of the descending colon [20].

Individuals with effective humoral immunological response and effective immunoglobulin G, upon encountering *C. difficile*, are more likely to become asymptomatic carriers of the bacterium [8]. Immunocompromised patients or patients under immunosuppressive therapy are at greater risk of developing the disease [8]. As CDI in immunocompromised patients is more severe, and colonization from infection is often difficult to distinguish, vigilance is needed to achieve early diagnosis [8]. A vast proportion of immunocompromised patients can become carriers of the bacterium.

CA-CDI has been related to inflammatory bowel disease in the literature. There is a thin line between colonization and infection in this group of patients, and the data are unclear [5,83]. In a retrospective analysis of patients with inflammatory bowel disease, 56% of the patients were diagnosed with CA-CDI, and the majority were not exposed to antibiotics [83]. Alteration in the gut microbiota in patients with inflammatory bowel disease is well-described in the literature. Chronic inflammation and a decreased, altered, and imbalanced gut microbiome can favor *C. difficile* colonization without antibiotic intake or hospitalization. Recent studies support this hypothesis as some data indicate that patients with inflammatory bowel disease become asymptomatic carriers of the bacterium at a higher rate than the control group [84,85]. Increasing evidence in the literature supports the idea that patients with inflammatory bowel disease suffer from or are carriers of *C. difficile* in the community [6,7]. These patients are younger than the rest of the carriers or patients with CDI [86]. Some researchers suggest that CDI could trigger the initial disease manifestation and favor relapses in IBD [87,88]. An imbalance of the gut microbiota in favor of facultative anaerobes is a common denominator for *C. difficile* colonization in both non-IBD and IBD patients as it facilitates the colonization of the bacterium [89]. Other predisposing factors for colonization in IBD are malnutrition, administration of antibiotics, biological agents, nonsteroidal anti-inflammatory drugs, and immunosuppressant therapy [90,91,92]. Administration of TNF-α inhibitors doubles the possibility of CDI in patients with IBD. The use of infliximab favors disease recurrence [93,94]. The impact of *C. difficile* colonization and possible infection in IBD patients has not been elucidated. More recent data support the notion that patients may have worse outcomes [95].

Patients suffering from chronic kidney disease or end-stage renal disease are more susceptible to severe CDI and recurrence [96]. Immune system impairment leads to an augmented risk for bacterial infections in patients with chronic kidney disease [97]. These patients show intestinal dysmotility and gut microbiota dysbiosis, are often hospitalized, and receive antibiotic treatment and gastric acid suppression; thus, the risk of colonization with nosocomial pathogens is increased [96,98]. Direct kidney injury due to *C. difficile* toxemia has been described in mouse models of oral toxin exposure. Injury is afflicted in a dose-dependent manner [99]. In humans, rare case studies pose a suspicion of acute renal injury due to CDI [100]. More studies are needed to elucidate possible direct kidney injury due to *C. difficile* infection [101,102]. The distinction between colonization and infection in these patients is difficult. Disease should be suspected in patients with changes in stool composition and odor and over three episodes of diarrhea in the same day [96]. They are exposed to antibiotic use and frequent hospitalization or visiting hospitals on an outpatient basis and show gut microbiota misbalance [103]. *C. difficile* is a significant pathogen in solid organ transplantation, leading to more severe disease and recurrences up to 40% [104,105]. CDI incidence is referred to be about 23% in lung transplant recipients [106]. Colonized hematology–oncology patients, both in the community and hospital settings, suffer from CDI at an incidence up to 33%. The incidence for human immunodeficiency virus (HIV) patients is also high [106]. Elderly patients admitted in geriatric wards are colonized by the bacterium at a rate up to 16% with toxigenic strains. It is suggested to screen these patients for *C. difficile* prior to admission as a preventive tool [98].

## 4. Most Common Ribotypes of *C. difficile* Related to CA-CDI

The high proportion of mobile genetic elements in the *C. difficile* genome renders the bacterium capable of surviving and adapting in demanding milieus [107].

There is no uniform method of molecularly typing isolates from community-acquired CDI. PCR ribotyping is the most used method in Europe to classify these strains [108].

Certain ribotypes related to hospital CDI were described early in the literature. RT009 was isolated in the United States in 1980, RT027 in France in 1985, RT017 in Belgium in 1995, RT017 in Ireland in 2006, and RT078 in the United Kingdom in 2007 [109,110,111,112]. RT027 was isolated worldwide from then on [107].

The first fully sequenced genome of isolated CDI in Zurich, Switzerland, was described in 2006 by Sebaihia and colleagues as the RT012 strain [113]. RT005 and RT020 were significantly associated with CA-CDI [114].

Whole-genome sequencing analysis of the most common ribotypes, RT002, RT005, RT023, RT020, and RT078, in Sweden revealed that isolates RT002 and RT078 showed the least variability via single-nucleotide polymorphism (SNP) analysis (median = 1 SNP and 5 SNPs, respectively). More than 30 SNPs were observed for RT005, which suggests association rather than sporadic cases of CA-CDI and outbreaks. A range of 3–30 SNPs were detected for isolates RT020 and RT023 [114].

CDI outbreaks in Europe and the United States have been linked to the highly virulent strains RT027 and RT078 [115,116].

The hypervirulent RT027 strain, or restriction endonuclease analysis type BI, North American pulsed-field gel electrophoresis type 1 (NAP1), or polymerase chain reaction (PCR) ribotype 027 (BI/NAP1/027), is the result of multiple genetic rearrangements. In the last 20 years, this strain has acquired five additional genetic regions compared to its wild type. In this way, it has adopted multiple means to survive and provoke severe disease. Its arsenal includes increased toxin production and binding, high-level fluoroquinolone resistance, and motility [117].

The RT027 strain possesses the genes that favor the production of binary toxins. In addition, certain deletions of the *tcdC* gene enhance the production of toxins A and B [118].

A survey conducted in China also revealed RT027 as the most virulent isolate in patients with CDI. In the same study, strains isolated from patients with diarrhea were also isolated from healthy individuals, which suggests that virulence strains circulate in the community. The carriage rate of healthy adults in the community was reported to be 5.5%. One third of the strains from healthy individuals were the ST54 isolate. This isolate is also related to CDI in Europe [119].

Ribotypes RT001, RT002, RT015, RT017, and RT018, related to CA-CDI, have been reported in the literature [120]. Ribotype 018 (RT018), the most common strain in Japan and Italy, was isolated from both community and hospital settings [121]. *C. difficile* RT017 is a toxin B producer and is related to outbreaks worldwide [122].

In descending order, primarily RT046, RT012, RT001, and RT009 were isolated in a study from China regarding patients with diarrhea with community-acquired characteristics [123].

RT023 is related to severe disease and CDI relapses [124].

CA-CDI has been related to RT106 and RT027 in many European countries since the late 2000s [125].

*C. difficile* RT027, RT078, and RT244 are binary toxin-positive strains and are related to epidemics worldwide [122].

## 5. Zoonotic Spread of *C. difficile* in the Community: The Role of Companion Animals in Cross-Contamination and Spread

The isolation of similar *C. difficile* toxigenic strains in both healthy individuals and animals supports the zoonotic spread of the bacterium in the community [108].

A study in China including hospital dogs and cats revealed they were carriers of *C. difficile*. All isolates from cats were toxigenic. Strains from dogs were toxigenic at a rate of 60%. RT027 was also isolated. The study’s results show that companion animals such as dogs and cats can be carriers of toxigenic strains of the bacterium, and cross-transmission can occur [126]. Isolation of RT027 from companion animals has also been reported in Canada [127,128].

*C. difficile* ribotype 078, related to severe human CDI, is a clone of the strain initially originated from swine and cattle and has now also been identified in retail meat [108].

RT014, a well-described culprit of human CDI in Europe, has also been isolated from companion animals. The isolation rate of RT014 in companion animals was 22.2% in a study from Germany [127,129,130]. Data from a study on companion animals in China reported RT014 and RT106 as the main pandemic strains. These isolates are also related to human CDI. The study’s authors concluded that more attention should be paid to companion animals and CDI for public health security [126].

Ribotypes 878, 879, 020, and 014 have been isolated from wastewater [120]. RT014 predominates in CDI in children [122]. Australian pigs are reservoirs of RT014 and RT020 [131]. In Iran, the hypervirulent RT078 has been isolated from wastewater [120]. The presence of the bacterium in wastewater favors community spread as wastewater may act as a reservoir [120].

In retail meat of veal, lamb, chicken, goat and turkey, a higher prevalence of the microorganism has been described in the USA [60]. *C. difficile* was first isolated from a goat in 1981 [132]. From thereon, there are many relevant reports globally regarding goats, sheep, and calves: in Nigeria [133], in Egypt [134], in Ireland [61], in Belgium [55], in India [135], in Slovenia [136], in Saudi Arabia [137], in Australia [138], and in Iran [139].

From 1996 onwards, *C. difficile* has been isolated from sheep, lamb and poultry in an age-related manner. The older the animal, the less the stool carriage detected both in farms and at slaughterhouses [55]. Another risk factor is the use of antibiotics as a common practice in farmed animals [140]. Regarding poultry, an outbreak of *C. difficile* symptomatic disease has been described in young ostriches [141]. Worldwide, poultry is known to be colonized by the bacterium. *C. difficile* has been isolated from stool cultures from broiler chickens [142] and retail chicken meats [62,63,64,65,66]. *C. difficile* was isolated from soil enriched with poultry-derived manure and could be detected for two years after soil fertilization [52].

## 6. Antibiotic Resistance of *C. difficile* Strains

Various studies from the last 25 years have demonstrated that *C. difficile* has acquired resistance genes to various antibiotics [107]. Antibiotic resistance in combination with toxin production is alarming, as they provoke more severe CDI [143,144]. Regarding the mobilome of *C. difficile*, antibiotic resistance is not plasmid-dependent but is mediated by transposable elements [107]. Transposable elements of *C. difficile* (transposons (Tns)) can be either mobilizable or conjugative. Mobilizable Tns depend on the host mechanisms, whereas conjugative transposons (CTns) or integrative and conjugative elements (ICEs) are self-transmissible [145,146,147]. Conjugative transposons, such as Tn916, have been reported as present in isolates with multidrug resistance related to epidemics [113,148,149].

Tn916, Tn5397, TnB1230, and Tn5398 are related to tetracycline resistance. Tn4453a/b is associated with chloramphenicol resistance. Tn5398 and Tn6215 are responsible for macrolide–lincosamide–streptogramin B resistance [107]. Tn6218, related to RT001, RT017, and RT078, is associated with reduced susceptibility to linezolid [150]. Tn6194, related to RT027 epidemics, contains *erm(B*) and is capable of intraspecies and interspecies gene transfer [107]. Bacteriophage φC2 of *C. difficile* mediates the transfer of Tn6215 containing *erm(B)* between two strains of *C. difficile* in vitro [151].

Mutations in the quinolone resistance-determining region of the *gyrA* and *gyrB* genes are related to resistance to fluoroquinolones [152,153]. Toxigenic isolates from CA-CDI with decreased susceptibility to moxifloxacin have been described early in the literature in Germany, related to mutations in *gyrA*. Some of these strains have been genotypically related to strains isolated from nosocomial settings [154].

CCD-1-like β-lactamase, related to carbapenem resistance, has been described in patients with severe disease [155]. QacG, related to severe CDI, is responsible for multidrug resistance [82]. *VanY* is responsible for glycopeptide resistance [155]. The allele *vanZ* reported in *C. difficile* isolates is related to teicoplanin resistance [125].

A strain of the hypervirulent isolate RT027 was described in a female patient suffering from CA-CDI. This isolate bore the *tet* gene, related to tetracycline resistance, and the *ermB* and *Tn916* genes, related to erythromycin/clindamycin resistance [156]. RT027 isolates with an MIC to vancomycin of >2 μg/mL have acquired the *vanGCd* gene cluster and *vanR* mutation [157,158]. RT002, RT014, and RT005 isolates containing the *VanS* mutation have been described [158].

Isolates demonstrating clindamycin resistance (with an MIC of >16 μg/mL) had the *ermB, ermQ*, and *ermG* genes. Most of them were RT027 [158]. Clindamycin-, florfenicol-, and chloramphenicol-resistant strains mostly belong to RT027 and possess the *cfrC*, *cfrB*, and *cfrE* genes [107,158].

*C. difficile* RT017 isolates have been reported to be resistant to fluoroquinolones and clindamycin [122].

Table 1 summarises the isolates of *C. difficile* related to CA-CDI worldwide and their characteristics.

## 7. Conclusions

The mobilome of *C. difficile* shows remarkable levels of plasticity. Genetic exchange between animals and humans is of concern. Although CA-CDI is not established as a food-borne or zoonotic disease, some data support this direction. No gold standard or ISO procedures have been developed for the detection of *C. difficile* in raw materials and food products in the food industry. The spores’ ability to survive standard cooking procedures and on abiotic surfaces, the formation of biofilms, and their survival within biofilms of other bacteria render even a low number of spores capable of food contamination and spreading. Meanwhile, adequate enumeration methods for the detection of a low number of spores in food have not yet been developed. Primary care physicians should take CA-CDI into consideration in the differential diagnosis of diarrhea. Surveillance is needed, as there is a thin line between colonization and infection. In patients diagnosed with inflammatory bowel disease and other comorbidities, *C. difficile* can be the cause of recurrent disease and should be included in the estimation of diarrhea and worsening colitis symptoms. In the community setting, it is difficult to distinguish asymptomatic carriage from true infection. For asymptomatic carriage, antibiotic therapy is not suggested but contact isolation and hand-washing practices are required. Primary healthcare providers should be vigilant and implement infection control policies for the prevention of *C. difficile* spread.

## Figures and Tables

**Table 1 ijms-26-04716-t001:** Isolates of *C. difficile* related to CA-CDI worldwide and their characteristics.

Ribotype of *C. difficile*	Characteristics
RT005 [114]	Responsible for sporadic cases; *vanS* mutation [114,158]
RT020 [114]	Responsible for outbreaks [114] Present in wastewater [158] Isolated from Australian pigs [131]
RT001 [123]	Reduced susceptibility to linezolid [150]
RT106 [125]	Isolated from companion animals [126]
RT017 [122]	Responsible for outbreaks [122] Related to food production [11] Reduced susceptibility to linezolid [150] Resistance to fluroquinolones and clindamycin [122]
RT078 [122]	Related to epidemics [114,122] Reduced susceptibility to linezolid [150] Isolated from swine, cattle, retail meat [108], wastewater [120], and food products [11]
RT244 [122]	Related to epidemics [122]
RT027 [122]	Related to epidemics and outbreaks [115,122] Resistance to erythromycin, clindamycin, and chloramphenicol; high-level resistance to fluoroquinolones; vancomycin MIC > 2 μg/mL; and resistance to tetracycline [107,156,157,158] Isolated from companion animals [126,127] and food products [11]

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
