# Peer review of "Community-Acquired Clostridioides difficile Infection: The Fox Among the Chickens"

_ijms, 2025, doi:10.3390/ijms26104716_

Round 1
Reviewer 1 Report
Comments and Suggestions for Authors
The authors presented a review of CA-CDI, including virulence factors, sources of colonization, CDI in IBD, role of gut microbiota, ribotypes, and antibiotic resistance. The encompassing coverage, although commendable, also result to lack of focus and granularity of content in each section. Would recommend to focus the paper more on Community-acquired C diff "colonization" and not "infection". Of note, the prevalence of CA-CDI is complicated by the challenges in diagnostic and antimicrobial stewardship in the community setting. Thus, what we may be calling as "infection" may actually be "colonization". However, CA-CDI truly exists and this starts with acquisition of the clostridial spores (i.e., colonization). Discussion more on environmental (food, water, animal) sources, molecular features of the strains (here the ribotypes and antimicrobial susceptibilities would be included) and how these may eventually get into the human host and potential implications (infection, transmission, recurrence, preventive measures), would be more relevant. There is, indeed, a "thin line" between colonization and infection but I do not think the paper actually addressed what makes for the crossing that line. There has been a lot written/reviewed about infection, including infection in the IBD population and thus, switching the focus more on "colonization" would give the article more impact.
Specific comments-
Virulence factors: only receptors for TcdA discussed. How about receptors for TcdB, since the later appears to be more relevant in human infection and the presence of TcdB+ but TcdA negative strains.
Virulence factors/diagnosis: molecular assays are more commonly used now, with or without EIAs, although this may be location specific (US vs other countries; resource limited vs sufficient settings, etc). Perhaps, good to discuss diagnosis in different/various context.
Lines 103-105: The statement here appears to imply that R027 is the only strain that is binary toxin positive, which is not accurate.
Means and sources of colonization: I think this is a really important section. However, this section extensively referred to reference number 8, which is also a review article. I strongly recommend that the authors review the original research articles, including updated information, rather than using the said reference mostly.
Conclusions: Instead of just mentioning CA-CDI as a differential diagnosis for diarrhea in patients presenting to their providers, this needs to be discussed in the appropriate context to avoid diagnosis of infection in a patient who is actually only colonized. This issue/point can be better discussed in a section focusing on the clinical implications of C diff colonization in the community setting.
Comments on the Quality of English Language
There are few spelling errors and the sentence construction may not be consistent with native speaker's style/structure.
Author Response
Dear Reviewer
Thank you for your comments. All suggestions have been included in this revised version of our manuscript.
All corrections are highlighted in yellow
Comment1) The authors presented a review of CA-CDI, including virulence factors, sources of colonization, CDI in IBD, role of gut microbiota, ribotypes, and antibiotic resistance. The encompassing coverage, although commendable, also result to lack of focus and granularity of content in each section. Would recommend to focus the paper more on Community-acquired C diff "colonization" and not "infection". Of note, the prevalence of CA-CDI is complicated by the challenges in diagnostic and antimicrobial stewardship in the community setting. Thus, what we may be calling as "infection" may actually be "colonization". However, CA-CDI truly exists and this starts with acquisition of the clostridial spores (i.e., colonization).
Responce 1) Text has been added in the revised manuscript to emphasize on colonization, lines 27-31, 47-55, 534-538
Comment 2) Discussion more on environmental (food, water, animal) sources, molecular features of the strains (here the ribotypes and antimicrobial susceptibilities would be included) and how these may eventually get into the human host and potential implications (infection, transmission, recurrence, preventive measures), would be more relevant. There is, indeed, a "thin line" between colonization and infection but I do not think the paper actually addressed what makes for the crossing that line. There has been a lot written/reviewed about infection, including infection in the IBD population and thus, switching the focus more on "colonization" would give the article more impact.
Response 2) Chapters 3 and 4 are combined in the revised manuscript under the title ‘‘Community spread in healthy individuals and immunosuppressed patients- The distinct role of gut microbiota’’. In this chapter other comorbidities except IBD are discussed, lines 340-346, 369-389 in the revised manuscript. Text has been added in the revised manuscript to emphasize on colonization, lines 27-31, 47-55, 534-538
Comment 3) Virulence factors: only receptors for TcdA discussed. How about receptors for TcdB, since the later appears to be more relevant in human infection and the presence of TcdB+ but TcdA negative strains.
Response 3) It has been added, lines 78-87, 92-101, 107-108, 153-156 in the revised manuscript
Comment 4) Virulence factors/diagnosis: molecular assays are more commonly used now, with or without EIAs, although this may be location specific (US vs other countries; resource limited vs sufficient settings, etc). Perhaps, good to discuss diagnosis in different/various context.
Response 4) It has been changed, lines 122-137 in the revised manuscript
Comment 5) Lines 103-105: The statement here appears to imply that R027 is the only strain that is binary toxin positive, which is not accurate.
Response 5) It has been changed, lines 158-159 in the revised manuscript.
Comment 6) Means and sources of colonization: I think this is a really important section. However, this section extensively referred to reference number 8, which is also a review article. I strongly recommend that the authors review the original research articles, including updated information, rather than using the said reference mostly.
Response 6) It has been changed, original research articles has been used and additions has been made, lines 237-306 in the revised manuscript.
Comment 7) Conclusions: Instead of just mentioning CA-CDI as a differential diagnosis for diarrhea in patients presenting to their providers, this needs to be discussed in the appropriate context to avoid diagnosis of infection in a patient who is actually only colonized. This issue/point can be better discussed in a section focusing on the clinical implications of C diff colonization in the community setting.
Response 7) It has been changed, lines 27-31, 47-55, 534-538, 340-346, 369-389 in the revised manuscript.
Comment 8) Comments on the Quality of English Language. There are few spelling errors and the sentence construction may not be consistent with native speaker's style/structure.
Response 8) The revised manuscript has undergone English language editing by MDPI. The text has been checked for correct use of grammar and common technical terms, and edited to a level suitable for reporting research in a scholarly journal.
Reviewer 2 Report
Comments and Suggestions for Authors
In section 2.1 the last paragraph is entitled "Biofilm formation". It only mentions what this property contributes to, but does not contain any information about how C. difficile forms a biofilm. I recommend to complete this missing information.
Chapters 3 and 4 overlap (3: IBD, 4: IBD and gut microbiota), I recommend combining them into one chapter and and creating a chapter in which you will introduce CA CDI and other comorbidities (chronic kidney disease, immunodeficiency and others).
Chapter 6 offers only brief information on zoonotic spread and does not include all animals (goats, sheep, poultry). Complete it.
In chapters 5-7 the text alignment on the right side is missing.
Author Response
Dear Reviewer
Thank you for your comments. All suggestions have been included in this revised version of our manuscript.
Comment 1) In section 2.1 the last paragraph is entitled "Biofilm formation". It only mentions what this property contributes to, but does not contain any information about how C. difficile forms a biofilm. I recommend to complete this missing information.
Response 1) It has been completed, lines 185-231 in the revised manuscript.
Comment 2) Chapters 3 and 4 overlap (3: IBD, 4: IBD and gut microbiota), I recommend combining them into one chapter and creating a chapter in which you will introduce CA CDI and other comorbidities (chronic kidney disease, immunodeficiency and others).
Response 2) Chapters 3 and 4 have been combined in the revised manuscript under the title ‘‘Community spread in healthy individuals and immunosuppressed patients- The distinct role of gut microbiota’’. In this chapter other comorbidities except IBD are discussed, lines 340-346, 369-389 in the revised manuscript.
Comment 3) Chapter 6 offers only brief information on zoonotic spread and does not include all animals (goats, sheep, poultry). Complete it.
Response 3) It has been added in the revised manuscript, lines 465-478
Comment 4) In chapters 5-7 the text alignment on the right side is missing.
Response 4) It has changed
Round 2
Reviewer 1 Report
Comments and Suggestions for Authors
This version is significantly better. Few additional suggestions/comments:
> Intro first sentence: "CDI mainly appears as nosocomial..." add that community acquired infection is increasingly being recognized.
> several places where "colonial" was used as a term. I think the authors meant "colonic"
> Line 103: change "bacterium" to "toxin". It is the toxin/s that get internalize via the receptors, not the bacteria.
> Line 319-321: The statement "The common risk factors..." is not accurate. Although patients with CA-CDI may be younger with less comorbidities, older age and antibiotic-induced dysbiosis remain significant factors for acquisition of true infection. This is one of reasons why it is important to differentiate colonization from true infection, especially in the community setting as it is likely that a significant portion of these "patients" are actually not infected.
> line 338: "lieu" should be "milieu"
Comments on the Quality of English Language
Author Response
Comment 1) Intro first sentence: "CDI mainly appears as nosocomial..." add that community acquired infection is increasingly being recognized.
Response 1) It has been added in the revised manuscript, lines 37-38.
Comment 2) several places where "colonial" was used as a term. I think the authors meant "colonic".
Response 2) It has changed throughout the revised manuscript.
Comment 3) Line 103: change "bacterium" to "toxin". It is the toxin/s that get internalized via the receptors, not the bacteria.
Response 3) It has been changed.
Comment 4) Line 319-321: The statement "The common risk factors..." is not accurate. Although patients with CA-CDI may be younger with less comorbidities, older age and antibiotic-induced dysbiosis remain significant factors for acquisition of true infection. This is one of reasons why it is important to differentiate colonization from true infection, especially in the community setting as it is likely that a significant portion of these "patients" are actually not infected.
Response 4) It has been changed, lines 318-322 in the revised manuscript
Comment 5) line 338: "lieu" should be "milieu"
Response 5) It has been changed.